# Clinical Trial for the Safety and Feasibility of Pedicle Screws Coated with a Fibroblast Growth Factor-2-Apatite Composite Layer for Posterior Cervical Fusion Surgery

**DOI:** 10.3390/jcm12030947

**Published:** 2023-01-26

**Authors:** Katsuya Nagashima, Yuki Hara, Hirotaka Mutsuzaki, Yasukazu Totoki, Eriko Okano, Kentaro Mataki, Yukei Matsumoto, Yohei Yanagisawa, Hiroshi Noguchi, Yu Sogo, Atsuo Ito, Masao Koda, Masashi Yamazaki

**Affiliations:** 1Department of Orthopedic Surgery, Faculty of Medicine, University of Tsukuba, 1-1-1 Tennodai, Tsukuba 305-8575, Japan; 2Department of Orthopedic Surgery, Ibaraki Prefectural University of Health Sciences, Ami 300-0394, Japan; 3Technology Research Institute, National Institute of Advanced Industrial Science and Technology, Tsukuba 305-8560, Japan

**Keywords:** ossification of the posterior longitudinal ligament, cervical spine, surgery, neck pain

## Abstract

To solve the instrument loosening problem, we developed a fibroblast growth factor-2-calcium phosphate composite layer as a novel coating material to improve screw fixation strength. The primary aim of the present study was to demonstrate the safety and feasibility of screws coated with the FGF-2-calcium phosphate composite layer for posterior instrumented surgery of the cervical spine. The trial design was a single-arm, open-label, safety and feasibility study. Patients receiving fusion of the cervical spine from C2 (or C3) to C7 (or T1) were recruited. The primary endpoint to confirm safety was any screw-related adverse events. Seven patients who underwent posterior fusion surgery of the cervical spine were enrolled in the present study. The coated pedicle screws were inserted bilaterally into the lowest instrumented vertebrae. There was only one severe adverse event unrelated with the coated screw. Three out of the fourteen coated screws showed loosening. The present results prove the safety and feasibility of pedicle screws coated with the FGF-2-calcium phosphate composite layer for fusion surgery in the cervical spine. This is the first step to apply this novel surface coating in the field of spine surgery.

## 1. Introduction

Recent advances in spinal instrumentation have improved clinical outcomes for various kinds of spinal pathologies. However, the increased usage of spinal instrumentation has led to the emergence of spinal instrumentation-specific surgical complications including increased risk of infection [1,2,3] instrumentation-related neuro-vascular injury [4,5,6,7,8] and loosening of implants [9,10,11]. Extensive efforts have been made to overcome these instrumentation-related surgical complications. Among these complications, implant loosening, which can lead to the failure of bony fusion potentially resulting in unfavorable clinical outcomes [12], is one emerging surgical complication, especially for older populations with osteoporosis in Japan [13].

To solve the instrument loosening problem, several countermeasures have been proposed. A combination with hook/wire [14] and penetrating endplate screws [15], both of which increase pull-out strength of the screws, has been used as techniques to prevent loosening. Other approaches include using materials, different screw configurations [16,17,18], cement augmentation [19], or various kinds of surface coating [20,21].

We developed an apatite basic fibroblast growth factor (Ap-FGF) coating device as a novel material to improve screw fixation strength [22]. It is composed of a surface coating of hydroxyapatite, which increases initial fixation strength, combined with basic FGF (bFGF) through co-precipitation, which can achieve the controlled release of bFGF and result in increased osteogenesis [23,24,25]. Although the FGF-apatite composite layer itself has an osteoconductive effect, the coated screw has no osteoconductive effects because the bonding between composite layer and screw surface is not so tight, as proven by the extraction test [23]. Therefore, the mechanism preventing screw loosening might promote bone formation around the screw without direct bonding, as shown in our previous basic animal study [23]. We previously performed a trial of Ap-FGF-coated external fixation pins for distal radius fractures in five patients that resulted in no severe adverse events, demonstrating safety and feasibility [26]. We propose that Ap-FGF-coated screws might prevent pedicle screws from loosening.

The primary aim of the present study was to demonstrate the safety and feasibility of Ap-FGF-coated screws for posterior instrumented surgery of the cervical spine.

## 2. Methods

The trial design was a single-arm, open-label, safety and feasibility study. The registration number was UMIN000026771 (https://center6.umin.ac.jp/cgi-open-bin/ctr/ctr_view.cgi?recptno=R000030735 (accessed on 1 August 2017)). Inclusion and exclusion criteria are shown in Table 1. Patients receiving long fusion of the cervical spine from C2 (or C3) to C7 (or T1) were recruited. The primary endpoint to confirm safety was any screw-related adverse events. Secondary endpoints were the rate of loosening/breakage of the Ap-FGF-coated pedicle screws and duration until bony fusion, to assess efficacy. The Japanese Orthopedic Association (JOA) score and visual analogue scale (VAS) for neck/arm pain were used to assess clinical outcomes.

### Trial Protocol

#### Ap-FGF Coating

An outline of the steps for processing the Ap-FGF coating is shown in Figure 1. Processing was performed according to good manufacturing practice (GMP) standards in a cell processing facility at our institute. Three screws were coated with Ap-FGF at one time, one for a quality check and two for surgery. The diameter and length of the pedicle screws were determined using a preoperative CT measurement. Titanium alloy pedicle screws were used in the present study (Mounteneer^®^, DePuy Synthes spine, Bethesda, MA, USA). bFGF (Fiblast^®^, Kaken Pharma, Tokyo, Japan; 4 μg/mL) was dissolved in an over-saturated phosphate–Ca solution (Otsuka Pharma, Tokyo, Japan) and screws were soaked for 48 h at 37 °C. A specific jig was made to hold the screws within the solution (Figure 2A). After 48 h of soaking, white precipitate on the bottom of the cup and coating on the surface of the screws were confirmed (Figure 2B,C).

Thickness of the composite layer was approximately 500 nm. Although we did not perform scratch tests, bonding between the screw surface and composite layer was not so strong because most of composite layer remained screw hole after screw extraction test [23]. By histological examination, biodegradation of composite layer showed no difference with apatite alone [23].

One out of three coated screws then served to check quality. The Ap-FGF-coated layer was dissolved in citrate solution and a protein assay (Bradford method), Ca/P concentration (elementary analysis) and bFGF bioassay (NIH 3T3 cell proliferation) were performed. The solution was collected and checked for endotoxins.

Eight patients who underwent posterior fusion surgery of the cervical spine were recruited into the present study (Figure 3 and Table 2). Indications for surgery included ossification of the posterior longitudinal ligament (6 cases) and cervical spondylotic myelopathy (2 cases). The average age at surgery was 59.6 years old (48–64 years old).

Ap-FGF-coated pedicle screws were inserted into the lower instrumented vertebrae. All the patients were followed by assessments of clinical outcomes, X-rays (~1 year after surgery), CT (1 week and 6 months after surgery), and laboratory investigations for 1 year after surgery. The definition of screw loosening was a result of observations of a lucent zone measuring 1 mm or more on the CT scan. Bony union was evaluated by continuity between facet joints on a sagittal reconstruction CT multiplanar image 6 months after surgery or motion of spinous process less than 2 mm in flexion–extension X-ray 1 year after surgery.

Adverse events were defined as any symptom or disease observed in a participant after informed consent with or without a causal relationship to the Ap-FGF screw. All adverse-event-related terminology was coded by the investigators according to the ICH International Medical Dictionary for Regulatory Activities, Japanese version (MedDRA/J). Those determinations were performed by an independent data and safety monitoring committee.

All procedures used in this study were approved by the institutional review board. Written informed consent was obtained from all patients to participate in this study and for publication. The datasets generated and analyzed during the current study are available from the corresponding author on reasonable request.

## 3. Results

One patient (No. 2) was excluded because of an incomplete procedure of the screw coating process (saline for the soaking solution had expired, Figure 3). Therefore, seven patients received Ap-FGF-coated pedicle screw insertion.

One patient (No. 4) died from a rupture of an abdominal aneurism six months after surgery. This was considered to have “no relationship with the Ap-FGF screw” (Table 3).

Other adverse events included laboratory data abnormalities, elevation of creatin phospho-kinase (seven cases), liver dysfunction (four cases) and elevation of serum amylase (one case), all of which were considered to have “no relationship with the Ap-FGF-coated screw” (Table 3). As for neurological adverse events, one case of postoperative C5 palsy occurred, which was also considered to have “no relationship with the Ap-FGF-coated screw” (Table 3). Other types of adverse events included pneumonia (one case) and urticaria (one case), both of which was also considered to have “no relationship with the Ap-FGF-coated screw” (Table 3). There was no surgical site infection or malignant diseases during the follow-up period.

Three out of fourteen Ap-FGF screws showed loosening. One of those was positioned incorrectly, possibly influencing loosening. All loosening was detected with follow-up CT six months after surgery (Figure 4 and Figure 5). Bony union was achieved in all patients one year after surgery. All clinical outcomes were favorable. The average JOA score improved from 10.3 points preoperatively to 15.3 points one year after surgery (*p* = 0.03). Postoperative VAS arm pain was significantly reduced (71.7 mm to 25.6 mm, *p* = 0.001), although there was no significant difference between pre- and postoperative neck pain (61 mm to 48 mm, *p* = 0.65).

## 4. Discussion

The present trial showed no adverse events associated with the Ap-FGF screws. The only severe adverse event (SAE) was death due to the rupture of an abdominal aortic aneurism. This SAE was considered to have “no relationship with the Ap-FGF-coated screw” because the patient died six months after surgery and a retrospective check of the preoperative CT revealed the aneurism.

Abnormalities detected by laboratory analyses were determined as “no relationship with the Ap-FGF-coated screw” because the number and degree of abnormalities was comparable to a standard postoperative course and rapidly resolved without any specific treatment. Therefore, they were possibly influenced by surgical invasiveness and/or drugs used in the perioperative period, including anesthetics, analgesics, and antibiotics, and the incidence of those postoperative laboratory abnormalities were similar in previous patients receiving instrumented fusion surgery of cervical spine. Those determinations were performed by independent data and safety monitoring committee. Previous reports revealed that bFGF is safe for the central nervous system because it had been used safely in laboratory investigations and clinical trials for spinal cord injury and brain infarction [27,28]. As for neurological adverse events, one case of postoperative C5 palsy occurred, which was considered to have “no relationship with the Ap-FGF-coated screw” (Table 3) because previous reports revealed incidence of C5 palsy after fusion surgery for cervical spine as approximately 10%, similar incidence in the present study [8].

Our previous report showed that screw loosening occurred up to 40% of the time at the caudal end of the long cervical fusion construct, possibly because long fusion leads to a longer lever arm [29]. Countermeasures for screw loosening at the end of the constructs are mandatory because mechanical stress concentrations can occur there. In the present study, 3 of 14 Ap-FGF pedicle screws inserted to the caudal end of the long fusion loosened. The rate of screw loosening was comparable to our previous series. However, we cannot draw a definitive conclusion because the present study was designed as a single-arm study without any control group. Further exploration using randomized controlled design is mandatory to elucidate the true efficacy of the Ap-FGF coating on the pedicle screw to counteract screw loosening.

## 5. Conclusions

The present results prove the safety and feasibility of Ap-FGF coating for pedicle screws for fusion surgery in the cervical spine. This is the first step to apply this novel surface coating in the field of spine surgery.

## Figures and Tables

**Figure 1 jcm-12-00947-f001:**
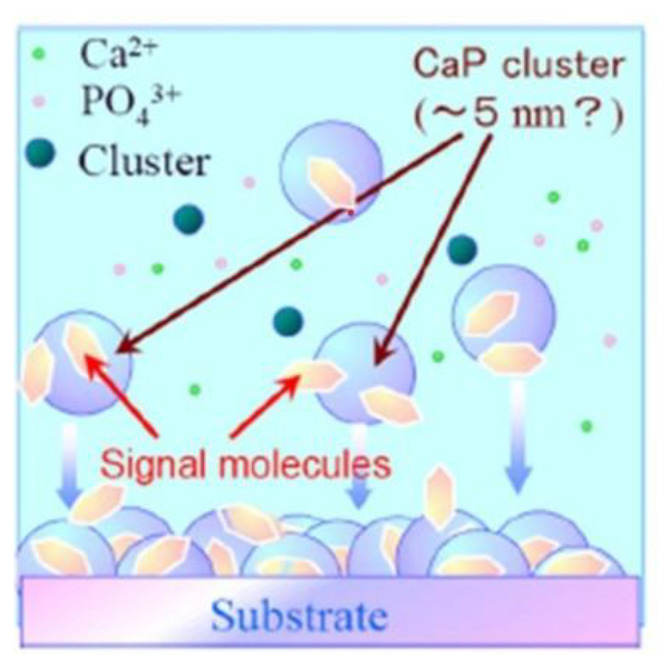
Outline of apatite basic fibroblast growth factor (bFGF) coating. It is composed of a surface coating of hydroxyapatite, which increases initial fixation strength, combined with bFGF through co-precipitation.

**Figure 2 jcm-12-00947-f002:**
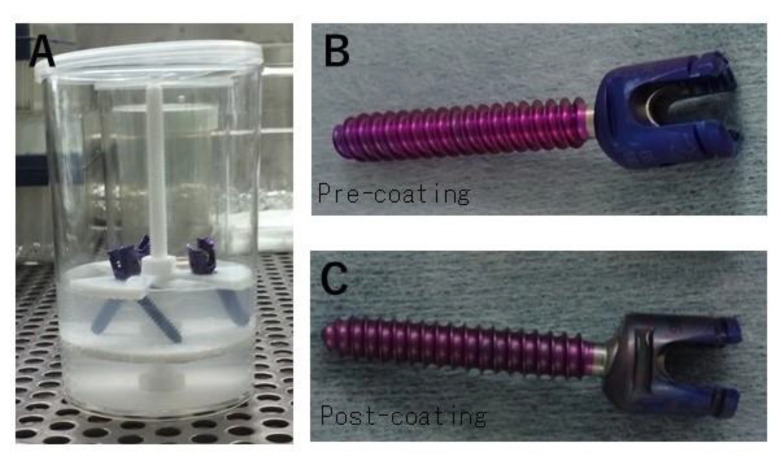
Apatite bFGF coating. Titanium alloy pedicle screws were used in the present study (Mounteneer^®^, DePuy Synthes spine, Bethesda, MA, USA). bFGF (Fiblast^®^, Kaken Pharma, Tokyo, Japan; 4 μg/mL) was dissolved in an over-saturated phosphate–Ca solution (Otsuka Pharma, Tokyo, Japan) and screws were soaked for 48 h at 37 °C. A specific jig was made to hold the screws within the solution (**A**). After 48 h of soaking, white precipitate on the bottom of the cup and coating on the surface of the screws were confirmed (**B**,**C**).

**Figure 3 jcm-12-00947-f003:**
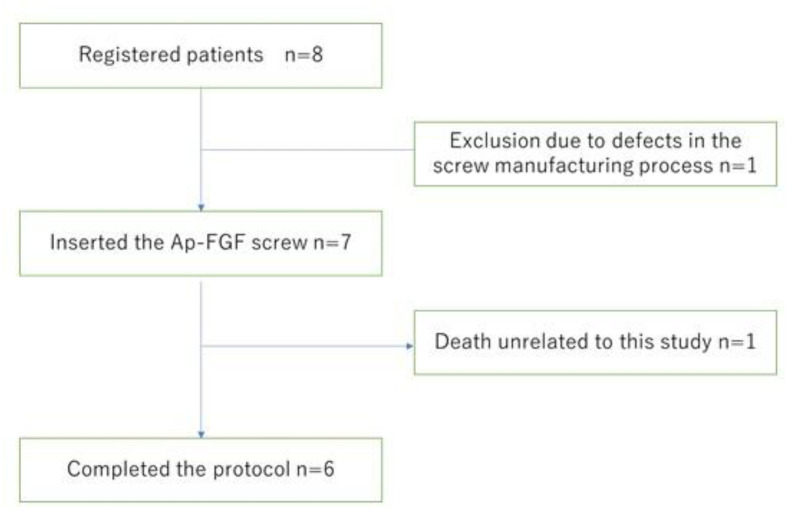
Patient recruitment flow.

**Figure 4 jcm-12-00947-f004:**
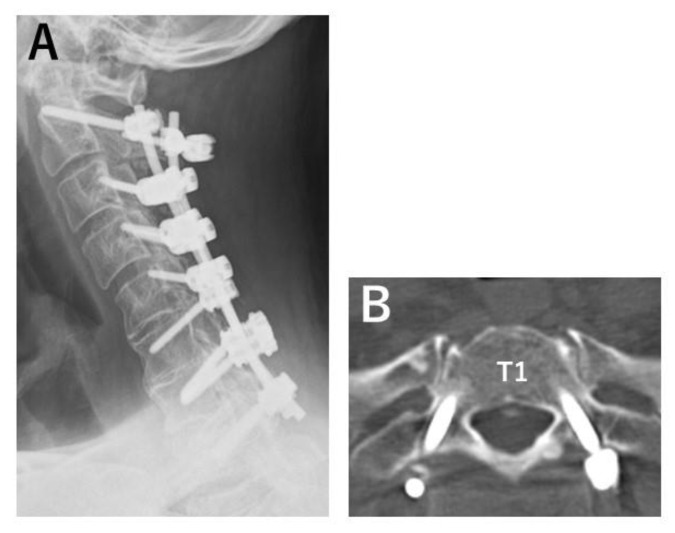
Postoperative images of a patient without screw loosening. Patient number 1. Apatite-bFGF-coated pedicle screws were inserted to T1 bilaterally. X-ray (**A**) and CT (**B**) obtained 6 months after surgery showed no lucent zone around pedicle screws.

**Figure 5 jcm-12-00947-f005:**
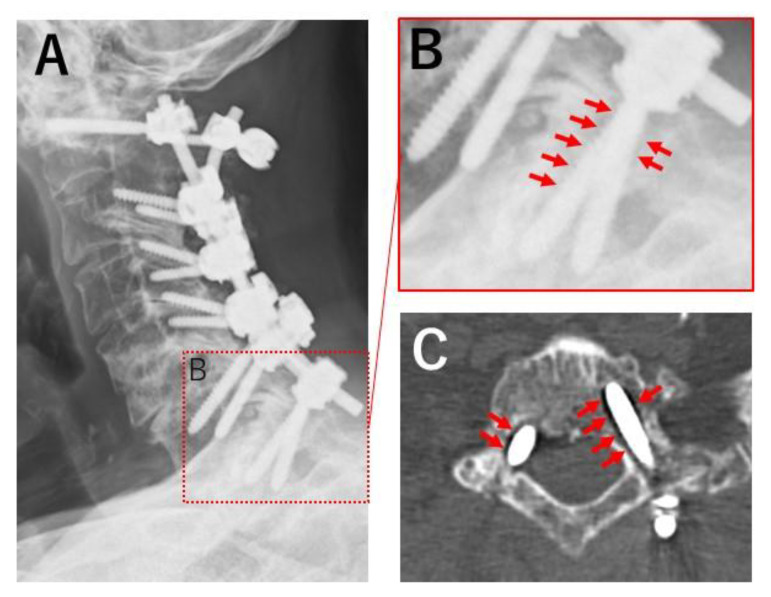
Postoperative images of a patient with screw loosening. Patient number 3. Apatite-bFGF-coated pedicle screws were inserted to T1 bilaterally. X-ray (**A**,**B**) and CT (**C**) obtained 6 months after surgery showed an apparent lucent zone around pedicle screws (**B**,**C**) arrows.

**Table 1 jcm-12-00947-t001:** Inclusion and exclusion criteria.

**Inclusion criteria** (1) Patients receiving long fusion (C2 or 3—C7 or T1) in cervical spine (2) Age from 20 to 90 years old
**Exclusion criteria** (1) History of cervical spine surgery, spondylitis or spinal tumor (2) Concomitant anterior cervical spine surgery (3) Included in other clinical trials within 3 months (4) Poor control of diabetes (5) Severe heart disease (6) Severe liver dysfunction (AST > 100 U or ALT > 100 U) (7) Severe renal dysfunction (BUN > 25 mg/dL or serum creatinine > 2.0 mg/dL) (8) Malignancy within 5 years (9) Pregnancy (10) Breast feeding (11) Allergy for titanium alloy (12) Possible allergy for basic FGF (13) Patients who have no ability for voluntary agreement (14) Patients who cannot be obtained written consent

**Table 2 jcm-12-00947-t002:** Patient demographics.

Pt. No.	Age	Sex	Main Disease	Comorbidities
1	59	M	OPLL	DM
2	61	M	OPLL	
3	61	M	OPLL	DM
4	64	M	CSM	RA
5	52	M	OPLL	DM, HT, HL
6	74	M	OPLL	CKD, HT
7	48	M	CSM	dystonia
8	58	M	OPLL	Liver dysfunction

DM: diabetes mellitus; RA: Rheumatic arthritis; HT: hypertension, HL: hyperlipidemia; CKD: chronic kidney disease.

**Table 3 jcm-12-00947-t003:** Adverse events.

Adverse Events (AE)	No. of Patients	Note
Severe AE		
Death	1	rapture of AAA
AE		
Laboratory data abnormalities		
CPK elevation	7	
Liver dysfunction	4	
Amylase elevation	1	
Neurological		
C5 palsy	1	recovered
Others		
Urticaria	1	
Pneumonia	1	
Gonitis purulenta	1	

## Data Availability

The data that support the findings of this study are available from the corresponding author, upon reasonable request.

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
