# Peer review of "Clinical Trial for the Safety and Feasibility of Pedicle Screws Coated with a Fibroblast Growth Factor-2-Apatite Composite Layer for Posterior Cervical Fusion Surgery"

_jcm, 2023, doi:10.3390/jcm12030947_

Round 1

Reviewer 1 Report

Thank you for the opportunity to read this interesting paper. The authors reported clinical use of PS coated with FGF-2 CP composite layer for posterior cervical spine fusion surgery. This material is promising for achieving solid screw fixation and preventing screw loosening. However, there are a few minor concerns to be revised.

<Minor comments>

・Do the screws have an osteoconductive effect? (Line 46-55)

How do the screws prevent loosening? By direct bonding to the bone, osteoconductive? Or by promoting bone formation around the screw without direct bonding? The authors must have investigated how the screw improve screw fixation strength by previous studies of basic experiments. Please describe briefly how to improve screw fixation strength.

・characteristics of the surface layer

Please briefly state the characteristics of the surface layer based on previous studies.

Thickness of the layer, scratch test, biodegradation, and so on.

・neck brace

Did the authors use a neck brace after surgery? Please describe.

・time point of CT (line 139-140)

The authors detected screw loosening at six months after surgery as described at line 139. At line 110, all patients were followed by CT one year after surgery. Did the authors check CT at six months after surgery for the suspicious cases of screw loosening? Or did all patients were check by CT at six months and one year after surgery? Please describe.

Author Response

Do the screws have an osteoconductive effect? (Line 46-55)

>>> Although FGF-apatite composite layer itself has osteoconductive effect, coated screw has no osteoconductive effects because the bonding between composite layer and screw surface is not so tight. Therefore, the mechanism preventing screw loosening might be promoting bone formation around the screw without direct bonding, shown in our previous basic animal study.

We added the explanation of mechanism to prevent screw loosening in Introduction section added our previous paper describing basic animal experiments as ref. 23 (Mutzuzaki, 2007).

Please briefly state the characteristics of the surface layer based on previous studies. Thickness of the layer, scratch test, biodegradation, and so on.

>>> Thickness of the composite layer was approximately 500 nm. Although we did not perform scratch test, bonding between screw surface and composite layer is not so strong because most of composite layer remained screw hole after screw extraction test (Mutsuzaki, 2007). By histological examination, biodegradation of composite layer showed no difference with apatite alone. We added those statements in Methods section.  

Did the authors use a neck brace after surgery? Please describe.

>>> We did not apply neck braces in all the patients. In usual clinical settings, we do not apply any external fixation for patients receiving instrumented fusion for cervical spine.

・time point of CT (line 139-140)

>>> We performed CT scan 6 months after surgery in all the patients. We correct the methods section (1year → 6 months).

Reviewer 2 Report

The authors reviewed a novel pedicle screw coated with FGF-2 calcium phosphate composite layer for the cervical spine. The manuscript is well-written.  I would like to clarify one point:

1. Abnormal lab data were stated in 12 instances, and the majority were due to CPK elevation. Yet this was judged to be "unrelated" without any explanation.  Can you please clarify on what basis this conclusion was reached, and how can the authors justify and prove none of these lab data abnormalities AEs were unrelated?

Author Response

 Abnormal lab data were stated in 12 instances, and the majority were due to CPK elevation. Yet this was judged to be "unrelated" without any explanation.  Can you please clarify on what basis this conclusion was reached, and how can the authors justify and prove none of these lab data abnormalities AEs were unrelated?

>>> Abnormalities detected by laboratory analyses were determined as "no relationship with the Ap-FGF coated screw" because they were possibly influenced by surgical invasiveness and/or drugs used in the perioperative period, including anesthetics, analgesics, and antibiotics, and the incidence of those postoperative laboratory abnormalities were similar in previous patient receiving instrumented fusion surgery of cervical spine. Those determination was performed independent data and safety monitoring committee. We added those statements in Methods section  and Discussion section .